# CTC, ctDNA, and Exosome in Thyroid Cancers: A Review

**DOI:** 10.3390/ijms241813767

**Published:** 2023-09-06

**Authors:** Wenwen Wang, Zhiyao Zheng, Jianyong Lei

**Affiliations:** 1Division of Thyroid Surgery, Department of General Surgery, West China Hospital, Sichuan University, Chengdu 610041, China; 2Department of Neurosurgery, Peking Union Medical College Hospital, Chinese Academy of Medical Sciences and Peking Union Medical College, Beijing 100730, China

**Keywords:** liquid biopsy, CTC, ctDNA, exosome, thyroid cancers

## Abstract

Thyroid cancer has become more common in recent years all around the world. Many issues still need to be urgently addressed in the diagnosis, treatment, and prognosis of thyroid cancer. Liquid biopsy (mainly circulating tumor DNA (ctDNA), circulating tumor cells (CTCs), and circulating exosomes) may provide a novel and ideal approach to solve these issues, allows us to assess the features of diseases more comprehensively, and has a function in a variety of malignancies. Recently, liquid biopsy has been shown to be critical in thyroid cancer diagnosis, treatment, and prognosis in numerous previous studies. In this review, by testing CTCs, ctDNA, and exosomes, we focus on the possible clinical role of liquid biopsy in thyroid cancer, including diagnostic and prognostic biomarkers and response to therapy. We briefly review how liquid biopsy components have progressed in thyroid cancer by consulting the existing public information. We also discuss the clinical potential of liquid biopsy in thyroid cancer and provide a reference for liquid biopsy research. Liquid biopsy has the potential to be a useful tool in the early detection, monitoring, or prediction of response to therapies and prognosis in thyroid cancer, with promising clinical applications.

## 1. Introduction

Thyroid cancer, the most prevalent endocrine cancer, has become more widespread in recent decades around the world [1]. There are many unanswered concerns about thyroid cancer diagnosis, treatment, and prognosis. Currently, the gold standard for thyroid cancer screening is fine-needle aspiration cytology (FNAC) [2]. However, as it is an intrusive procedure that relies heavily on the operator’s abilities, approximately 15–30% of FNAC falls into the indeterminate category [3]. And, FNAC has limited diagnostic value for follicular thyroid carcinoma (FTC) [4]. Moreover, serum thyroglobulin (Tg) is a vital tool in identifying thyroid cancer recurrence after total thyroidectomy [5,6]. However, for the measurement of Tg, varying results are easily caused by the use of different methods [7,8]. Tg measurement is generally hampered by serum Tg antibody (TgAb) in the blood and reduces the accuracy of Tg as a predictor of differentiated thyroid cancer (DTC) activity [9,10]. In terms of therapy, radioactive iodine (RAI) therapy is considered a standard treatment method for most advanced DTC [11,12]. However, some DTCs develop into radioiodine refractory during treatment, and radioiodine-refractory thyroid carcinoma with distant metastases has a bad prognosis, with a median survival duration of 2.5 to 3.5 years [13]. RAI therapy has very limited benefits for patients with radioiodine-refractory DTC, who may receive unnecessary treatment [14]. Therefore, assessment and determination of the intensity of therapy and the avoidance of overtreatment are critical for radioiodine-refractory DTC patients. Liquid biopsy may predict the treatment effectiveness of RAI therapy. Therefore, the development of a more sensitive and reliable tumor marker that can dynamically reflect tumor genetic information and treatment response is critical to developing novel Radioactive Iodine 131(^131^I) therapeutic strategies. Liquid biopsy may provide new insight into solving these tasks (Figure 1).

Liquid biopsy, as a noninvasive, easy, and accessible detection method for tumor cells or tumor-derived products in body fluids, has developed dramatically in recent years [15]. It can detect the presence of genetic material generated from tumors, including circulating tumor cells (CTCs), circulating tumor DNA (ctDNA), exosomes, and others such as circulating free-DNA(cf-DNA) and circulating free-RNA(cf-RNA) as well as in peripheral blood, saliva, urine, and cerebrospinal fluid [16]. Compared with cf-RNA and cf-DNA, CTCs, ctDNA, and exosomes are more stable and specific [17]; so were three principal components of liquid biopsy in clinical application. In our review, we focus on CTCs, ctDNA, and exosomes.

Compared with tumor biopsies, liquid biopsies are more convenient and minimally invasive, which enables repetitive sampling and decreases the financial costs and potential complications of tissue biopsies [18]. In addition, the results of liquid biopsy detection can guide adjustments of treatment plans. Recently, liquid biopsy technology’s clinical applications in the early diagnosis, prognosis, and therapy of cancers, including prostate [19,20], lung [21,22,23], pancreatic [24], and colorectal cancers [25,26], have been actively investigated. Thyroid cancer is one of the most common endocrine tumors, and liquid biopsy’s importance in thyroid cancer has been widely discussed. There is now a large body of research on liquid biopsy for thyroid cancer.

Considering this and given the above-highlighted aspects, this article gives a review of how liquid biopsy components have progressed in thyroid cancer. We also discuss the clinical potential of liquid biopsy in thyroid cancer and provide a reference for liquid biopsy research (Figure 2).

## 2. CTC and Its Clinical Application

### 2.1. CTC

CTCs are malignant cells originating from both primary and metastatic tumors that are shed into the bloodstream or lymphatic ducts of cancer patients [27]. In 1869, the pathologist Ashworth et al. [28] first reported CTCs in the circulation of a deceased patient. Recently, many studies have shown a relationship between CTC levels and the development of cancers (e.g., breast cancer [29], colorectal cancer [30], and neuroendocrine cancer [31]). These results suggested that CTCs have the potential to be an effective tool for cancer diagnosis, informing clinical decision-making and clinical research.

The potential applications of enumeration and epithelial-mesenchymal transition (EMT) are far-reaching and play a critical role in CTCs. The majority of studies looking at CTCs in malignancies find a direct link between higher CTC counts and poor clinical outcomes [32]. CTC enumeration could be a well-known cancer diagnosis, prognosis, disease progression, and therapeutic response prediction biomarker [33]. In the sampled blood of cancer patients with breast cancer [34] and small cell lung cancer [21], the increasing numbers of CTC clusters were found to correlate with significantly reduced progression-free survival rates. In addition, EMT, which refers to how epithelial cells become interstitial cells after losing their polarity, is considered the most essential mechanism of tumor metastasis [35]. It is typified by the epithelial marker being downregulated and mesenchymal markers being upregulated [36]. EMT may play a critical role in CTCs, and several studies have demonstrated that EMT contributes to cancer progression by promoting CTC migration and invasion into vessels [37]. Additionally, the acquisition of EMT status in CTCs may lead to the expression of tumor stem cell characteristics and augment chemoresistance [38]. EMT may be a critical mechanism by which CTCs survive and obtain metastatic abilities.

A key technical challenge to the application is how to collect and separate the extremely uncommon CTC population from billions of background cells with great efficiency (typically one single CTC in 106–107 white blood cells from cancer patients’ peripheral blood) [39]. In recent years, CTC isolation and detection systems have been created in a variety of ways, relying on CTCs’ physical and biological qualities (Figure 3). Based on physical properties, CTCs are distinguished from blood cells by differences in cell size, density, and deformability [40,41,42]. Frequently used methods include density gradient centrifugation, membrane filtration, and electrophoretic separation (a physical method for separating tumor cells based on their size and polarity in an electric field), and each method has pros and cons [40,43,44,45]. The physics-based enrichment method separates CTC molecules by size and deformability. This method is simple, economical, and easy to mass produce, but it requires relatively large volumes of blood and has low capture efficiency and sensitivity [46]. Immunoaffinity-based enrichment refers to the technique that uses antibodies binding to surface markers of cells, including an anti-epithelia cell adhesion molecule (EpCAM), a tumor-specific cell surface antigen, cytokeratin (CK), and other stem cell or mesenchymal markers, and they work on the basis of two strategies: positive enrichment and negative enrichment [40,44]. The CellSearch method is the only one that has been authorized by the FDA to monitor CTC numbers in blood samples [45]. Positive selection using EpCAM-coated ferromagnetic particles and CTC enumeration are used to enrich CTCs [47]. This method is semiautomated and has superior sensitivity. However, several problems affect CTC identification, such as time and labor costs [43]. More importantly, it is possible that this approach may result in erroneous positive and negative results, which are related to changes in tumor markers during EMT [48]. Upstream immunomagnetic depletion is used to remove white blood cells in negative immunoaffinity-based enrichment [49]. Although the technology does not rely on the expression of surface biomarkers, the low purity also affects the practicality and limits its development [40] (Table 1).

### 2.2. CTCs in Thyroid Cancer

Since 1998, when Ringel et al. [50] confirmed CTCs in the blood of individuals with thyroid cancer and local recurrence or distant metastases, the involvement of CTCs in thyroid cancer has gained the attention of an increasing number of researchers (Table 2).

According to ample evidence, many individuals with malignant illnesses have CTCs, even in the early stage [51,52]. Therefore, the detection of CTCs could provide new opportunities for cancer diagnosis. FTC is mainly characterized by a follicular structure, and diagnostic controversy and interobserver variability make its diagnosis the most difficult aspect of thyroid pathology [53,54]. According to Sato et al. [55], positive CTC in the blood was more common in FTC patients, while benign patients were all negative (with 100% specificity and 46% sensitivity), revealing that CTC detection by RT-PCR may commonly discriminate between malignant and benign FTC. Moreover, Badulescu et al. [56] measured CTCs in 22 cases of minimally invasive follicular thyroid cancer (MIFC) and 4 cases of benign diseases. Patients who tested positive for CTCs had bigger tumors, more multifocality, and vascular invasion, suggesting that CTC testing is feasible regarding MIFC or benign thyroid tumors with a follicular pattern in individuals.

In addition, CTC detection has the potential to anticipate metastases and offer precise prognostic data. The tumor–node–metastasis (TNM) staging system is the main cancer staging system and provides basic guidance for disease prognosis. Among studies on other cancers, such as liver cancer, gastric cancer, and adrenocortical tumors, we observed a correlation between CTCs and T stage [57,58,59]. There are also reports of an association with thyroid carcinoma: according to Ehlers et al. [60], thyroid cancer patients had a much higher CTC level and the number was significantly increased with the increase in tumor T stage, indicating that the T stage of DTC patients may be a good independent factor correlated with the number of CTCs, and the number of CTCs was unaffected by DTC metastatic dissemination and lymph node involvement when compared to T stage. Qiu et al. [61] also demonstrated this viewpoint: to collect and detect CTC in DTC patients, they employed chromosome 8 negative enriching immunofluorescence in situ hybridization (NE-iFISH). The number of CTCs was substantially greater in DTC patients with distant metastases (DM) than that in the healthy comparison group. They discovered that CTCs ≥ 5 were strongly linked to DM+ DTC. Xu et al. [62] used the CellSearch system to detect the CTCs number in the peripheral blood of metastatic medullary thyroid cancer (MTC) and found that patients with CTCs ≥ 5 had a median survival duration of 51.5 months, while that of patients with CTCs ≥ 5 was only 13 months. Patients with metastatic MTC who have greater than or equivalent to 5 CTCs have a worse overall survival rate (OS).

Qiu et al. [61] detected CTCs in 72 DTC patients and 30 healthy controls before and after ^131^I therapy and found that CTCs ≥ 7 were linked to a poor ^131^I therapy response, suggesting that in DM+ DTC, CTCs might indicate a poor response to ^131^I treatment and a bad prognosis. Additionally, Winkens et al. [63] identified CTCs in DTC patients before radioiodine treatment and 2 days, 14 days, and 3 months following treatment. The finding revealed that the decrease in the amount of EpCAM+ circulating epithelial cells in the early stages of RAI therapy helped to identify patients who were sensitive to RAI treatment and suggested that CTCs may help ensure patients’ response to ^131^I therapy. The sodium/iodide symporter (NIS) expression of CTCs was observed by Zheng et al. [64], who discovered that reduced or stable total NIS+ CTCs following RAI treatment might suggest beneficial effectiveness or successful RAI therapy. Of note, the number of CTCs may be linked to the duration of radioiodine treatment, and the data show that compared to those < 8 years ago, CTCs are more common in DTC patients who received RAI therapy more than 8 years ago [60], which was the first study revealing the association between CTCs and the period since the last RAI therapy.

**Table 2 ijms-24-13767-t002:** Key studies investigating circulating tumor cells (CTCs) as biomarkers in thyroid carcinoma.

Study	Year	Technique of Detection	Study Population	Main Finding	Potential Clinical Applications	Ref
Sato et al.	2005	RT-PCR test platform	121 patients (44 patients with benign tumors, 77 patients with malignant tumors)	- CTC can often distinguish malignant from benign FTC	- Differentiates between benign and malignant follicular nodules	[55]
Xu et al.	2016	Cell Search	42 patients (18 metMTC, 14 metDTC, 9 DTC, 1 MTC)	- CTCs ≥ 5 in patients with metMTC is associated with worse overall survival	- Predicts prognosis in advanced TC	[62]
Badulescu et al.	2018	Immunomagnetic separation	26 patients (22 MIFCs, 4 benign thyroid tumors)	- CTC-positive patients may have larger tumors and more frequent multifocality and vascular invasion	- Diagnosis and staging of FTC	[56]
Qiu. et al.	2018	NE-iFISH	102 patients (72 DTC, 30 healthy controls)	- CTCs ≥ 5 to be a potential predictive index for DM+ DTC - CTCs ≥ 7 as a possible indicator of poor response to 131I treatment and worse prognosis in DM+ DTC	- Predicts metastasis and prognosis of DTC	[61]
Ehlers et al.	2018	Cell Search	67 patients (33 PTC, 20 FTC, 14 MTC)	- CTCs correlated to the T stage and the time point of radioiodine therapy	- Guides tumor staging judgment and RAI	[60]
Zheng et al.	2019	CanpatrolTM	234 DTC patients (all had undergone total thyroidectomy, 197 received one RAI therapy after total thyroidectomy)	- The numbers of CTCs after RAI therapy were definitely lower than baseline	- Monitor the efficacy of RAI	[64]

DM: distant metastases; MIFCs: minimally invasive follicular thyroid carcinomas; NE-iFISH: negative enriching immunofluorescence in situ hybridization.

Taken together, it is clear from the studies mentioned above that CTCs may be a novel biomarker to evaluate for guiding decisions regarding the diagnosis of thyroid cancer, choice of ^131^I treatment, and predicting patient prognosis. However, there are some shortcomings in these studies. First, most of these studies are retrospective studies, with fewer prospective, multicenter studies, and they include small sample sizes and shorter follow-up times. Second, the sampling for CTCs in most of these studies was single-site sampling, such as only preoperative sampling or postoperative sampling, which would not reflect the correlation between CTCs and different stages of thyroid cancer. Finally, the current studies are relatively scarce in exploring standardized CTCs for the diagnosis and treatment of different subtypes of thyroid cancer.

## 3. ctDNA and Its Clinical Applications

### 3.1. ctDNA

ctDNA represents tumor-derived DNA fragments found in body fluids including urine, blood, and cerebrospinal fluid [65]. ctDNA, accounting for a tiny percentage of cell-free DNA (cfDNA), is a kind of DNA that is obtained from and found nearly exclusively in tumor patients. It can be released from tumor cells into blood by multiple mechanisms, such as apoptosis, necrosis, and secretion [66,67]. The genetic changes in ctDNA are the same as those in the initial tumor, including loss of heterozygosity, mutation, methylation, and copy number alterations [68,69], which renders it a possible cancer-associated biomarker for diagnosis. Additionally, the level of ctDNA is influenced by disease severity and varies according to the disease burden, site, and tumor biologic characteristics [70]. Recent technological advancements have elevated ctDNA molecular analysis to one of the most potential future diagnostic techniques [71]. Previous research has shown that detecting ctDNA aids in the diagnosis and monitoring of cancer patients [72,73,74,75]. Moreover, ctDNA has a short half-life (less than two hours), making real-time molecular monitoring of cancer possible [76]. Moreover, rather than biopsy from only one region, such as puncture, ctDNA can capture inter- and intratumor heterogeneity [77]. These findings suggest that ctDNA may have exquisite biological specificity as a biomarker of cancer.

The amount of ctDNA in the circulatory system is relatively small, so the analysis of ctDNA requires highly sensitive and specific techniques. In recent years, the detection strategies of ctDNA have become increasingly mature. Several techniques are available for the analysis of ctDNA, including concentration-based detection and structure-based detection (mutation, methylation). According to the detection principle, it can be divided into two types: selective amplification and sequencing. The former includes real-time quantitative polymerase chain reaction (qPCR) and digital droplet PCR (ddPCR) [78,79,80,81]. They have high analytical sensitivity for mutation detection and have emerged as powerful tools for the detection of ctDNA, particularly for rare molecule detection [82]. However, its disadvantage is that there are just a few known mutations that can be recognized, and the method is low-throughput [83]. The latter refers to Sanger sequencing and next-generation sequencing (NGS [84]). Each method has its advantages and disadvantages. Sanger sequencing provides a number of benefits, including reliable detection, cheap cost, and a high success rate, but low throughput cannot meet the current demand [85,86]. The advent of NGS has enabled the generation of large amounts of sequence data, marking the start of a new era in genomics research. Compared with Sanger, NGS can produce millions of reads in a single run, but the efficiency and practicality can limit it in the clinic [87,88,89] (Table 1).

### 3.2. ctDNA in Thyroid Cancer

To date, the function of ctDNA in thyroid cancer has only been studied in a few research studies. Although the role of ctDNA in thyroid cancers is not yet defined, there is evidence of its potential applications (Table 3).

Chung et al. [90] discovered that BRAF mutations were detectable in the circulating DNA of 21% of PTC patients, whereas benign patients tested negative for the gene. Cradic et al. [91] also found that circulating BRAF^T1799A^ was detectable in the blood of PTC patients but not in non-PTC patients, suggesting that ctDNA could be a potential diagnostic marker in PTC patients. This idea was also backed by a prior study that provided a case report of circulating BRAF^T1799A^ in thyroid cancer [92]. Lan et al. [93] discovered that the ctDNA detection rate in PTC with DM was substantially higher than in PTC without DM, and the detection rate was associated with invasiveness and tumor size. Li et al. [94] found 44.07% of thyroid tumors were BRAF-V600E ctDNA-positive (with sensitivity 61.54%, specificity 90.91%), suggesting that the application of ctDNA to thyroid nodule detection is critical. Recently, a study by Sandulache et al. [95] demonstrated a 100 percent correlation between BRAFV600E identification in primary tumors and plasmids. In addition, Allin et al. [96] detected ctDNA in 67% of patients and the sensitivity was 76%. Additionally, in comparison to traditional markers, ctDNA levels appeared to fluctuate more quickly in response to changes in the sickness state, suggesting that ctDNA might be a therapeutically useful biomarker in thyroid cancer.

The methylation of ctDNA correlated with tumor-specific alterations and is considered a promising biomarker in several cancers. Currently, in several research studies, the hypermethylation of cytosine-phosphate-uracil (CpG) island promoters has been considered to be a crucial component in the inactivation of tumor suppressor genes in thyroid cancer [97,98]. In Khatami’s study [99], in PTC patients, the methylation of two O6-methylguanine-DNA methyltransferase (MGMT) promoter areas (c) and (d) was considerably higher than in goiter controls, suggesting that MGMT may be associated with PTC. Additionally, in 2013, Zane et al. [100] found that in thyroid cancer patients, hypermethylation of the SLC5A8 and SLC26A4 genes can help in diagnosis. In addition, Hu et al. [101] analyzed serum DNA methylation indicators in 39 thyroid cancer patients who had previously been treated, and 70% were positive among 10 patients proven to have recurrent disease by conventional measures. They suggested that ctDNA may be a novel tool for thyroid cancer recurrence monitoring.

However, when it comes to detecting ctDNA in patients with thyroid cancers, there is some debate. Although ctDNA holds great potential as a diagnostic biomarker, its clinical application is limited in thyroid cancer. Condello et al. [102] took plasma samples from thyroid nodule patients having surgery and patients with PTC, but in the blood of 22 individuals with the mutation, they were unable to discover peripheral BRAFV600E mutations, even using two kinds of highly sensitive techniques (real-time PCR and digital PCR). Zane et al. [100], additionally, found no BRAF mutations in 46 BRAF-positive PTC patients’ ctDNA. Moreover, Suh et al. [103] investigated BRAF, KRAS, NRAS, and TERT promoter mutations in ctDNA from 73 matched neoplastic tissue and plasma DNA samples, as well as 54 healthy patients’ plasma DNA samples. In early-stage thyroid tumors, there was no discernible link between tumor tissue mutations and plasma mutational data. In a study by Kwak et al. [104], using real-time PCR, 94 blood samples from PTC patients who had a BRAF mutation in their tumor were tested for ctDNA. However, BRAF mutations were not found in any of the patient’s serum. Lupo et al. [105] also did not recommend the use of ctDNA readings in conjunction with FNAC and molecular tests to diagnose PTC.

Through current studies, we know that ctDNA is one of the biomarkers with great potential for clinical application in the diagnosis and treatment of thyroid cancer. However, ctDNA is low in peripheral blood and difficult to extract, and it is unstable and susceptible to degradation by various enzymes. The application of ctDNA in the diagnosis and treatment of thyroid cancer is still in the stage of experimental research and has not been tested in clinical practice. In addition, studies on the role of ctNDA in thyroid cancer also have the limitations of small sample size, short follow-up time, and lack of large-scale multicenter and prospective studies.

**Table 3 ijms-24-13767-t003:** Key studies investigating circulating tumor DNA(ctDNA) as biomarkers in thyroid carcinoma.

Study	Year	Technique	Study Population	Main Finding	Potential Clinical Applications	Ref
Hu et al.	2006	Real-time PCR	39 previously treated thyroid cancer patients (10 had recurrent disease, 20 had no current disease)	- 70% of recurrent patients had positive serum DNA methylation indices	- A new tool for monitoring TC recurrence	[101]
Cradic et al.	2009	Allele-specific real-time PCR	193 patients (173 PTC, 8 FTC, 11 Hurthle cell carcinoma, 1 MTC)	- BRAFT1799A can be detected in the blood of PTC patients with residual or metastatic disease and may provide diagnostic information	- Diagnosis and follow-up of PTC	[91]
Chuang et al.	2010	Gap-ligase chain reaction technique	28 patients (14 benign disorders, 14 PTC)	- The BRAF mutation existed in tumor DNA samples obtained from PTC patients	- Diagnosis and prognostic judgment of PTC	[90]
Kwak et al.	2013	Real-time PCR	94 patients (67 PTMC, 27 lymphocytic thyroid	- None of the patients had a detectable serum BRAF(V600E) mutation.	--	[104]
Zane et al.	2013	Real-time PCR	181 patients (9 ATC, 58 MTC, 5 SMFC, 23 FA, 86 PTC)	- SLC5A8 and SLC26A4 hypermethylation appeared to be easy, reproducible, and non-invasive for the diagnosis of TC	- Non-invasive tools for the diagnosis of TC	[100]
Sandulache et al.	2017	Digital PCR	23 patients with ATC	- ctDNA testing may be considered the primary method of characterizing ATC genomic information	- Guides the treatment of patients with ATC	[95]
Lupo et al.	2018	PCR	66 patients with thyroid nodules	- ctDNA mutations were not sensitive or specific enough to provide valuable information over FNAB	--	[105]
Allin et al.	2018	Digital droplet PCR	51 patients (17 PTC, 15 FTC, 15 MTC, 3 poorly differentiated thyroid cancer, 1 anaplastic	- ctDNA can be detected in blood in the early stages of TC, including PTC, ATC, and MTC	- Diagnosis of TC at an early stage and assessment of treatment outcomes	[96]
Condello et al.	2018	Real-time PCR and digital PCR	83 patients (70 thyroid nodules undergoing surgery, 13 DM-PTC	- The corresponding ctDNA was negative by using both techniques	--	[102]
Khatami et al.	2019	MS-HRM	102 patients (57 PTC, 45 Goiter)	- Among 7 candidate regions of ctDNA, MGMT (c) and MGMT (d) showed higher sensitivity and specificity for PTC and were suitable candidates as biomarkers of PTC	- Diagnosis of PTC	[99]
Li et al.	2019	QuantStudio^TM^ 3D digital PCR	59 patients with PTC	- BRAF was the most common genetic mutation in PTC patients; 44.07% of patients were BRAF V600E-positive and associated with tumor aggressiveness	- Helps identify benign and malignant thyroid nodules, predict PTC prognosis, and track PTC progression	[94]
Lan et al.	2020	PCR	36 patients (10 N0M0, 11 N1M0, 15 DM)	- The detection rate of ctDNA was significantly associated with metastasis, invasiveness, and tumor size	- Guides risk stratification of patients with PTC	[93]
Suh et al.	2021	PNA-mediated real-time PCR	73 matched neoplastic tissues and plasma samples (62 TC, 8 benign thyroid disorders, 3 parathyroid lesions); 54 plasma samples from healthy individuals	- The clinical utility of BRAF, KRAS, NRAS, and TERT promoter mutation analysis on ctDNA appeared to be limited to early-stage thyroid cancers.	--	[103]

PNA: peptide nucleic acid; PCR: polymerase chain reaction; MS-HRM: methylation-sensitive high-resolution melting; SMFC: synchronous medullary and follicular thyroid cancers; FA: follicular adenomas.

## 4. Exosomes and Their Clinical Applications

### 4.1. Exosomes

Exosomes, tiny membrane vesicles that have a lipid bilayer spheroid structure, are secreted by the majority of cells [106]. Exosome formation is a fine-tuned process, including initiation, endocytosis, multivesicular body formation, and exosome secretion [107]. Studies have shown that exosomes can be found in a number of bodily fluids, including blood, saliva, urine, and other bodily fluids, and their roles are determined by the cell’s kind and origin. Moreover, due to their small size, exosomes can easily pass through cell barriers. They are also highly stable in a variety of body fluids, suggesting that they are ideal sources of biomarkers for clinical analysis [108].

More recently, exosomes are key components of the tumor microenvironment, according to numerous studies, and play a role in cancer progression [109]. Exosomes released by tumor cells can induce angiogenesis, increase vascular permeability, increase resistance, and direct immune escape, all of which are favorable conditions for carcinogenesis and cancer progression [110]. Exosomes are intimately linked with EMT, which is crucial for cancer metastasis to occur. Moreover, exosomes can be employed to transport medications for tumor-targeted therapy [107]. Therefore, the exploitation of the function of exosomes will provide potential clinical applications for tumor diagnosis and treatment.

Thus far, exosomes have been isolated by many methods, including differential ultracentrifugation, size-based isolation, immunomagnetic separation, and microfluidics [111]. Each method of isolation has inherent advantages and disadvantages. When it comes to exosome separation, the gold standard is ultracentrifugation, but it is also limited by time constraints, inefficient separation, clogged filter wells, and compromised exosome integrity [112]. Moreover, size-based isolation is more efficient and widely used in clinics. However, these methods face unique challenges, such as the clogging of filters and interference from impurity proteins [113]. Immunoaffinity-based methods can selectively separate exosomes, and they exhibit the merits of high specificity and recoveries. Nonspecific absorption may produce a minority of proteins, making it very difficult to generalize [114,115] (Table 1). 

### 4.2. Exosomes in Thyroid Cancer

#### 4.2.1. Exosomes as Biomarkers for Disease Staging and Diagnosis

Exosomes have been demonstrated to be useful as biomarkers in the diagnosis of thyroid cancer in a number of investigations (Table 4). Capriglione et al. [116] assessed exosomal miRNA expression in PTC patients and healthy controls; they discovered that in the serum of PTC patients, the levels of exosomal miR181a-5p, miR24-3p, miR382-5p, and miR146a-5p were considerably downregulated relative to the controls. Liang et al. [117] evaluated exosomal miRNA in PTC patients, benign thyroid nodule patients, and healthy people, and, compared with healthy controls, nodule patients had significantly lower levels of miR-223-3p, miR-34-5p, miR182-5p, and miR-146b-5p. PTC patients had significantly lower levels of serum exosomal miR-29a than healthy persons, as discovered by Wen et al. [118]. In addition, PTC patients had higher levels of serum exosomal miR-223-5p and miR-16-2-3p than nodular patients. Wang et al. [119] also found that in PTC plasma exosomes, miR-34a-5p, miR-346, and miR-10a-5p were all elevated and suggested that PTC can be distinguished from healthy individuals and nodular goiter by miRNA, which has significant clinical value. Other studies have shown that plasma exosomal miR-485-3p, miR-4433a-5p, and miR-5189-3p could be utilized as biomarkers to diagnose PTC [120,121]. Moreover, in a study by Smasonov et al. [122], the diagnostic utility of exosomal miRNAs was investigated in a large group of individuals with various thyroid disease statuses, such as PTC, FTC, and benign tumors. The research results revealed that the serum exosomes miR-31 level in PTC was dramatically increased and that exosomal miR-21 was increased in FTC. In addition, with 100% sensitivity and 77% specificity, miRNA-181a-5p and miRNA-21 were found to be useful in distinguishing between PTC and FTC. Other studies [123] have also pointed to miR-145 as a promising diagnosis marker for the invasion of PTC. Additionally, Chen et al. [124] discovered that in PTC patients with lymph node metastasis (LNM), the expression of miR-6879-5p and miR-6774-3p was significantly greater. Additionally, Jiang et al. [125] confirmed that in PTC patients, for predicting LNM, exosomal miR-222-3p and miR-146b-5p have a substantial diagnostic value. As a result, these exosomal miRNAs could be employed as PTC and FTC diagnostic markers.

Long non-coding RNAs (lncRNAs) are crucial factors in tumor microenvironments. Hardin et al. [126] demonstrated that exosomes can cause epithelial-mesenchymal transition (EMT), significantly upregulate the expression of lncRNA MALAT1 and linc-ROR, and induce a tumor microenvironment. Dai et al. [127] also found that exosomal lncRNA DOCK9-AS2 was shown to be elevated in PTC patients’ plasma and aggravate cancer progression by the Wnt/β-catenin pathway. In addition, SNHG9 was identified as a PTC cell exosome-enriched lncRNA by Wen et al. [128]. CircRNAs are a type of endogenous RNA that is covalently closed. They are extremely stable due to their unique structural properties. CircRNAs are regarded as innovative predictive biomarkers with clinical promise because they are protected by exosomes. A study by Yang et al. [129] identified changed exosomal circRNAs in the serum of a PTC patient using high-throughput sequencing and matched their findings to those of patients who had benign nodules. Circ_031752, circ_007293, and circ_020135 were upregulated in PTC, showing that the circulation of serum exosome RNAs could be used as PTC diagnostic molecular biomarkers. Lin et al. [130] also confirmed that circ_007293 was found to be abundant in exosomes from PTC patients, suggesting that it may influence the miR-653-5p/PAX6 axis to induce EMT. Wu et al. [131] discovered that circRASSF2 was elevated in PTC patients’ serum exosomes, and the expression level has a significant inverse correlation with exosomal miR-1178. The miR-1178/TLR4 pathway increased exosomal circFNDC3B expression in the serum of PTC patients, suggesting that it could be a helpful biomarker for PTC131.

In addition, Li et al. [132] provided evidence that in thyroid cancer, exosomal protein ANXA1 was overexpressed, which promoted cancer cell invasion and proliferation. Proteomic techniques were used to investigate the differently expressed proteins in the exosomes of PTC patients by Luo et al. [133] and then the exosome proteomes of PTC patients with LNM, those without LNM, and healthy volunteers were compared. They revealed that in PTC patients with LNM serum exosomes, integrin-associated proteins, such as SRC, TLN1, ITGB2, ACTB, and CAPNS1, were clearly upregulated, which suggested that PTC is associated with distinct changes in protein signaling. In addition, another study [134] assessed the level of plasma exosomes in PTC and benign goiter patients before and after ablative surgery. They revealed that PTC had higher levels of exosomal Hsp90, Hsp60, and Hsp27 than benign goiter, and they were also increased in PTC before and after ablative surgery, suggesting their potential as biomarkers for clinical applications. Luo et al. [133] also showed that in PTC patients with LNM’s exosomes, Hsp27 was abnormally overexpressed. Several studies [135,136,137] have shown that Galectin-3 is correlated with thyroid cancer, but its specific mechanism is still unknown.

**Table 4 ijms-24-13767-t004:** Key studies investigating exosomes as biomarkers in thyroid carcinoma.

Study	Year	Components	Technique	Study Population	Main Finding	Potential Clinical Applications	Ref
miRNA							
Boufraqech et al.	2014	miR-145	qRT-PCR	--	- Circulating miR-145 levels were significantly higher in patients with PTC	- Guides the diagnosis and targeted therapy of TC	[123]
Lee et al.	2015	miR-146b, miR-222	qRT-PCR	--	- miR-146b and miR-222 were demonstrated to be abundant in the exosome of PTC, causing a negative proliferation effect	- Determines recurrence of PTC	[138]
Samsonov et al.	2016	miR-21, miR-31, mri-181a-5p	qRT-PCR	50 patients (34 PTC, 8 FTC, 8 benign thyroid adenoma)	- miR-31 in the serum exosome of PTC was significantly upregulated and miR-21 increased in the FTC. miRNA-21 and miRNA-81a-5p were found to be expressed reciprocally in the exosomes of PTC and FTC	- Identify the benign and malignant nature of thyroid nodules	[122]
Wang et al.	2019	miR-346, miR-34a-5p, miR-10a-5p	qRT-PCR	3 PTC pools, 1 heathy control pool	- miR-346, miR-34a-5p, and miR-10a-5p upregulated in PTC plasma exosomes	- Guides the diagnosis of PTC	[119]
Jiang et al.	2020	miR-146b-5p, miR-222-3p	qRT-PCR	64 patients (49 patients with LNM, 15 patients without LNM)	- Plasma exosomal miR-146b-5p and miR-222-3p could serve as potential biomarkers for LNM in PTC.	- Determines if TC has lymph node metastasis	[125]
Pan et al.	2020	miR-5189-3p	qRT-PCR	20 patients (13 PTC, 7 nodular goiter)	- miR-5189-3p might serve as biomarkers for PTC diagnosis	- Identify the benign and malignant nature of thyroid nodules	[120]
Dai et al.	2020	miR-485-3p, miR-4433a-5p	qRT-PCR	252 patients (119 PTC, 82 benign thyroid nodules, 51 healthy controls)	- Significantly increased miR-485-3p and miR-4433a-5p expression in PTC patients	- Guides the diagnosis of PTC	[121]
Wen et al.	2021	miR-29a	qPT-PCR	219 patients (119 PTC, 100 healthy controls)	- Exosomal miR-29a levels were significantly downregulated in PTC patients compared to healthy people - Exosomal miR-29a may be associated with PTC recurrence	- Assists in the diagnosis and prognosis of PTC	[118]
Chen et al.	2021	miR-6774-3p, miR-6879-5p	qRT-PCR	Testing studies: 9 patients (5 PTC-N1, 4 PTC-N0); Validation studies: 59 patients (29 PTC-N1, 30 PTC-N0)	- miR-6774-3p and miR-6879-5p may serve as new promising biomarkers for the diagnosis of LNM in PTC patients	- Determines if TC has LNM	[124]
Capriglione et al.	2022	miR24-3p, miR181a-5p, miR146a-5p, miR382-5p	TaqMan Advanced miRNA Array Cards	Two different series of 56 and 58 PTC patients, 18 healthy controls	- Exosomal miR24-3p, miR181a-5p, miR146a-5p, and miR382-5p were significantly downregulated in the resum of PTC patients relative to that of others	- Assists in the diagnosis of PTC and LNM	[116]
Liang et al.	2020	miR-34-5p, miR182-5p, miR-223-3p, miR-146b-5p, miR-16-2-3p, miR-223-5p	qRT-PCR	96 patients (35 PTC patients, 30 benign thyroid nodules, 31 healthy controls)	- Compared to healthy controls, miR-34-5p, miR182-5p, miR-223-3p, and miR-146b-5p were significantly lower in nodules (*p* < 0.0001) - The level of exosomal miR-16-2-3p and miR-223-5p was higher in PTC patients than in nodule patients	- Identify the benign and malignant nature of thyroid nodules	[117]
IncRNA							
Hardin et al.	2018	lncRNA MALAT1, lncRNA ROR	qRT-PCR	--	- Exosomes can induce EMT in normal non-cancerous thyroid cells by the transfer and expression of IncRNA MALAT1 and IncRNA ROR	- Guides the treatment of ATC	[126]
Dai et al.	2020	lncRNA DOCK9-AS2	qRT-PCR	54 PTC patients and 44 healthy controls	- Exosomal lncRNA DOCK9-AS2 was upregulated in the plasma of PTC patients and aggravated cancer progression by the Wnt/β-catenin pathway	- As a therapeutic target for PTC	[127]
Wen et al.	2021	lncSNHG9	qRT-PCR	--	- SNHG9 was identified as a PTC cell exosome-enriched lncRNA	- Guides the treatment of PTC	[128]
circRNA							
Yang et al.	2019	circ_007293, circ_031752, circ_020135	qRT-PCR and high-throughput sequencing	3 PTC, 3 benign thyroid goiter	- circ_007293, circ_031752, and circ_020135 have been shown to be differentially expressed in the plasma of patients with PTC	- Guides the diagnosis, treatment, and prognosis of PTC	[129]
Wu et al.	2020	circRASSF2	qRT-PCR	10 patients (5 PTC, 5 healthy controls)	- circRASSF2 was upregulated in serum exosomes from PTC patients	- Guides the treatment of PTC	[131]
Lin et al.	2021	Circ_007293	qRT-PCR	80 patients (40 PTC, 40 healthy controls)	- circ_007293 was enriched in exosomes derived from PTC patients	- Develops new therapeutic strategies	[130]
**Proteins**							
Luo et al.	2018	SRC, TLN1, ITGB2, CAPNS1, ACTB, Hsp27	Western blot	49 patients (16 PTC with LNM, 17 PTC without LNM, 16 healthy controls)	- These proteins were clearly upregulated in the serum exosomes of PTC patients with LNM	- Assists in the diagnosis of PTC and LNM	[133]
Caruso et al.	2019	Hsp27, Hsp60, and Hsp90	Western blot	Benign goiter and PTC	- Hsp27, Hsp60, and Hsp90 were higher in PTC than in benign goiter	- Guides the diagnosis, treatment, and prognosis of PTC	[134]
Li et al.	2021	ANXA1	Western blot	3 PTC patients	- Exosomal protein ANXA1 was overexpressed in thyroid cancer and increased the proliferation and invasion of cancer cells	- Develops new therapeutic strategies	[132]

lncRNA: long non-coding RNAs; LNM: lymph node metastasis.

#### 4.2.2. Exosomes as Biomarkers for Metastasis and Recurrence Monitoring

The timely supervision of thyroid cancer recurrence is critical. Exosomes have been shown to perform a role in thyroid cancer metastasis and recurrence in studies. Wen et al. [118] found that in patients with PTC who had a high level of serum exosomal miR-29a, the OS and RFS were considerably greater than in individuals with lower exosomal miR-29a expression, demonstrating that the levels of exosomal miR-29a may be associated with PTC recurrence. As Lee et al. reported [138], miR-146b and miR-222 were demonstrated to be abundant in the exosomes of PTC, causing a negative proliferation effect, which can potentially be a marker of PTC recurrence. In addition, Huang et al. [139] enrolled 16 PTC and FTC patients and collected their urine exosomal proteins such as thyroglobulin and galectin-3 before, during, and after the procedure, as well as three and six months later. In this study, they discovered an increase in urine exosomal Tg and radioactive ^131^I ablation in five patients after surgery, but no increase in serum Tg, revealing that the increasing tendency of urinary exosomal thyroglobulin may suggest probable recurrence.

In summary, exosomes also play an important role in liquid biopsy of thyroid cancer and have great potential for clinical application in the diagnosis, treatment, and prognosis of thyroid cancer. However, the current studies on exosomes in thyroid cancer also have many shortcomings, such as insufficient sample size and no control between different ethnic populations; additionally, the extraction and purification of exosomes need to be further optimized. Also, these studies do not suggest which exosome biomarker is the most valuable in the diagnosis and treatment of thyroid cancer. Previous studies have found that exosomal miRNAs can cause Hashimoto’s thyroiditis by influencing cellular communication pathways [140]. But, the mechanisms by which exosomes regulate tumor development and their biological functions in vivo also need to be further investigated.

## 5. Future Perspectives

With the development of technology and multidisciplinary convergence in recent years, the application of liquid biopsy has shown promising advancement in thyroid cancer, including earlier evaluation, monitoring of disease status, evaluating the efficacy of therapeutic intervention, and so on. Alternatively, several studies have shown that artificial exosome mimetics have the ability to be substitutes for exosome-based drug delivery [141]. The characteristics of exosomes, such as biocompatibility, small particle size, and low immunogenicity, offer great opportunities to expand access to develop the application of exosomes as a kind of potential medication delivery system in various cancers [142,143]. A new study [144] looked at the capacity of exosomes produced from thyroid cancer to target original tumors using a newly developed bioluminescent reporter system. In addition, Wang et al. [145] found that ATC (anaplastic thyroid carcinoma) proliferation was reduced and cellular apoptosis was promoted by exosomes containing SCD-1 siRNA. These results indicated that liquid biopsy may possess great potential as a viable choice for thyroid cancer treatment. Interestingly, single-cell profiling techniques, including single-cell sequencing and cytometry, provide an unprecedented opportunity to research the heterogeneity of the immune infiltration of tumors [146]. Currently, several studies have combined liquid biopsy and single-cell sequencing analyses, finding that they may be able to provide more information for a variety of cancer diagnoses and classifications [147,148,149]. However, thyroid cancer has not yet been investigated. This may be an important direction for future research.

Tumor status is often monitored after TC by biochemical markers such as thyroglobulin (Tg) and calcitonin (CT), as well as imaging tests such as ultrasound and computed tomography. However, there are deficiencies in these monitoring tools. Biochemical indicators such as serum Tg may be affected by Tg antibodies (TgAbs) [150]. In addition, Tg has limited clinical value in postoperative TC follow-up for poorly differentiated thyroid cancer [151]. CT levels can be affected by other diseases such as chronic renal failure and parathyroid disease [152]. Imaging examinations may have adverse effects on the human body due to repeated radiation exposure. Therefore, there is a need to develop new methods for tumor surveillance. Liquid biopsy allows for molecular characterization with tumor heterogeneity and representativeness via body fluids [17] and is expected to be a powerful tool in the diagnostic and therapeutic process of TC, guiding the preoperative, intraoperative, and postoperative management of TC. Liquid biopsies may be more useful in combination with currently available clinical testing tools. Among CTC, ctDNA, and exosomes, exosomes have more potential in liquid biopsy due to their relatively high content, richer biological information, and high biological stability [153]. We believe that the combination of two or three markers may be more promising for utilizing liquid biopsy in the promotion of thyroid cancer.

Currently, liquid biopsy has been studied more in lung cancer and breast cancer among solid tumors, and its detection technology is relatively mature and has been applied in their clinical diagnosis and treatment [154]. However, there are still many hurdles and practical concerns to be addressed for the use of liquid biopsy biomarkers in thyroid cancer clinical trials. Because the number of liquid biopsy components, such as CTCs, ctDNA, and exosomes, is limited and lacks standardization between the detection methodology and confounding factors, liquid biopsy faces technical challenges [18,155,156]. First, the separation and purification techniques associated with liquid biopsies need to be optimized, and the detection techniques need to be further simplified. In addition, the costly and time-consuming nature of liquid biopsy testing is a major obstacle to the popularization and application of liquid biopsy in clinical practice. Currently, potential markers derived from different studies are also not uniform and may differ by ethnicity and disease subtype. As mentioned above, there is still much debate about the role of ctDNA in thyroid cancer.

Moreover, other liquid biopsy biomarkers, in addition to CTCs, ctDNA, and exosomes, could be as important to investigate. Studies have shown that using multiple biomarkers in combination to predict cancer holds more predictive power than using single biomarkers alone. In thyroid cancer, the combination of two exosomal miRNAs in plasma helps to diagnose lymph node metastasis [124]. Similar reports have been made in other cancers. For example, in the diagnosis and treatment of primary lung cancer [157], breast cancer [158], and colon cancer [159], the combination of multiple biomarkers may be more conducive to early diagnosis, precision treatment, and prognosis assessment of these cancers than a single biomarker. Therefore, how to use all of these biomarkers together is also a question that needs to be considered in diagnosing and treating thyroid cancer. In addition, all studies had a relatively small number of participants. To define and validate the role of liquid biopsy, larger and more robust research is needed.

This review is intended to provide a comprehensive overview of liquid biopsy in the field of thyroid cancer. We analyzed a substantial number of studies as an update of existing studies. However, this review also has some limitations. First, there was a difference in the quality of the studies, the patients participating in the study, and the types of interventions in each study. Second, few studies have been performed to evaluate specific molecular mechanisms. Future studies should focus on the mechanisms of liquid biopsy in thyroid cancers. Finally, the data from the included studies were limited to published studies. Therefore, our review will be updated in the future to include more studies.

## 6. Conclusions

With the continuous development of new technologies, we have a better understanding of the formation mechanism, biological function, and relationship with thyroid cancer of liquid biopsies, such as CTCs, ctDNA, and exosomes. They have good potential for application in the diagnosis prognosis of thyroid cancer. Liquid biopsy is projected to become a novel cancer diagnostic, monitoring, prognostic, and treatment response prediction marker.

## Figures and Tables

**Figure 1 ijms-24-13767-f001:**
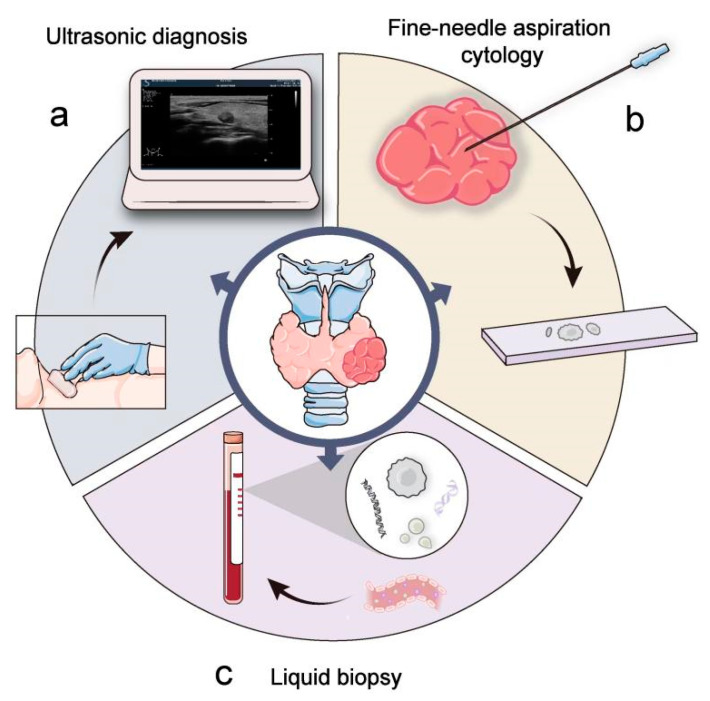
Clinical examination and diagnosis of thyroid cancer. Ultrasound (**a**) and fine-needle aspiration biopsy histology (**b**) are commonly used methods for detection and diagnosis of thyroid cancer. Liquid biopsy (**c**) offers a new approach to diagnosis.

**Figure 2 ijms-24-13767-f002:**
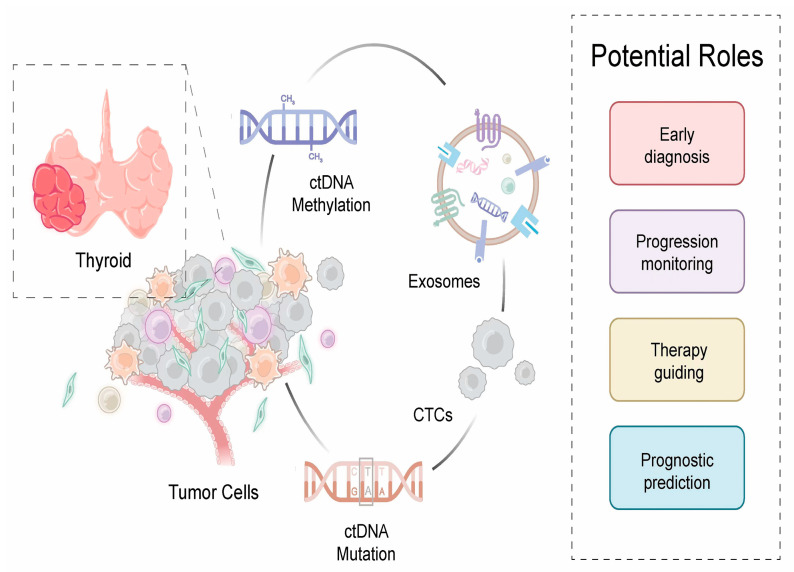
Potential clinical application of liquid biopsy in thyroid cancer. Liquid biopsy, including circulation tumor DNA (ctDNA), circulating tumor cells (CTCs), and exosomes, could be a potential tool for early diagnosis, effective monitoring, precise treatment, and prognostic prediction.

**Figure 3 ijms-24-13767-f003:**
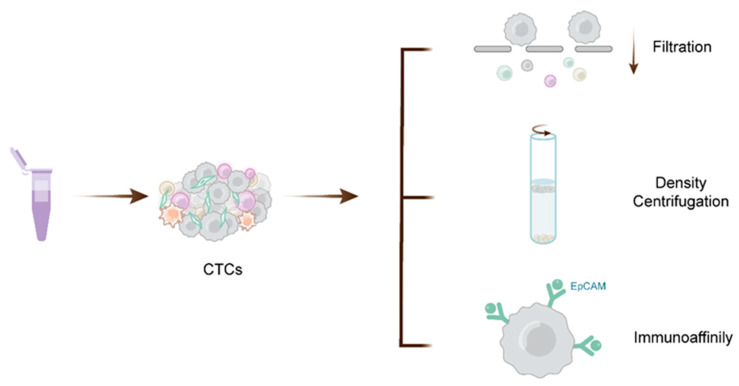
Isolation and detection methods for CTCs. Numerous technologies have been developed for CTC detection relying on the physical and biological properties of CTCs. Physical methods mainly include separation and enrichment by filters or filter membranes based on CTC size and gradient centrifugation and filtration utilizing the density of CTCs. Immunoaffinity is a method of enrichment using specific antigens expressed on the surface of CTCs, such as EpCAM.

**Table 1 ijms-24-13767-t001:** Methods for isolation and analysis of liquid biopsy components. Overview of the advantages and disadvantages of the described methods.

LB Component	Technique	Advantages	Disadvantages
CTCs	Physical-based enrichment	- economic - simple and easy to mass produce - quick processing time	- require relatively large volumes of blood - low capture efficiency and sensitivity
	Immunomagnetic-based enrichment	- semi-automated - superior sensitivity	- remaining time and labor costly - false positive and false negative finding
ctDNA	qPCR	- cheap - high specificity	- a limited number of known mutation can be detected
	ddPCR	- absolute quantity - high sensitivity	- low-throughput
	Sanger sequencing	- reliable detection - low cost - high success rate	- low throughput
	NGS	- high throughput - rich information	- limited by the efficiency and practicality
exosomes	Differential ultracentrifugation	- cheap	- the risk of contamination - low throughout
	Size-based isolation	- more efficient - environmentally friendly - widely used in clinic	- the clogging of filters - interferences of impurity protein
	Immunomagnetic separation	- high specificity, - high recoveries	- produce impurity protein

qPCR: real-time quantitative polymerase chain reaction; ddPCR: digital droplet PCR; NGS: next-generation sequencing.

## Data Availability

No new data were created or analyzed in this study. Data sharing not applicable to this article.

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
