# Peer review of "CTC, ctDNA, and Exosome in Thyroid Cancers: A Review"

_ijms, 2023, doi:10.3390/ijms241813767_

Round 1
Reviewer 1 Report
The authors provide a comprehensive overview of an important and evolvling medical-scientific field: the application of liquid biopsy in thyroid cancer patients. The manuscript is well written and inforative, with only very few typing errors, which easily can be removed by minimal editing. Figures are good, but the Table 1 should include separating lines between CTCs, ctDNA and exosomes to avoid confusing the readers.
English is good. There are only a couple of easily findable spelling errors, but not affecting the content.
Reviewer 2 Report
The manuscript is clear, well detailed and structured. In this review, the authors well described the clinical potential applications of liquid biopsy in thyroid cancer patients. However, authors must provide more details regarding several aspects of their literature search.
· The authors should give more space to the conclusion and the weaknesses rather than the purpose and results. Thus, authors should consider inserting a “limitation section” in the paragraph of the results. In this section, authors could report the weaknesses reported in each single study citated and also their overall considerations on this aspect.
· How the authors explain the failure of diagnostic applications of liquid biopsy to date?
· This review article provides an excellent topical overview on the potential application of liquid biopsy in the diagnosis of thyroid cancer. However, authors should emphasize the potential application of liquid biopsy in the diagnosis rather than during the follow-up of thyroid cancer patients. Please in table 2, other to the main finding column, authors should add another column regarding the clinical application (in the diagnosis, or in follow-up, or as indicator of therapy response, of survival and of prognosis ..). These data could help the readers to understand the likely future scenarios in this field.
· What is the future development and application of liquid biopsy? What are the considerations of the authors regarding the clinical application of CTC, ctDNA, and exosome in thyroid cancer? Which are the most promising markers among them?
· Consider the different subtypes of thyroid cancer (papillary, follicular, undifferentiated or medullary thyroid cancer), authors should add more details concerning the application of liquid biopsy among these different subtypes. It might be interesting for the readers to have a Table that highlight in which thyroid tumor subtype each study was conducted.
· The authors should better specify the study populations both in Table 2 and in the text (age, gender, thyroid cancer stage ..). This could help the readers to better comprise the potential application of liquid biopsy and to compare the studies citated in this review among them.
· The application of CTC, ctDNA, and exosome need to be reworking focusing on findings, placing them in the context of other known biomarker actually used in the management of thyroid cancer patients (e.g., Tg, TgAb, TSH…). This information is necessary to highlight eventual correlation among CTC, ctDNA, and exosome and the biomarker used in the clinic.
· In the papers citated are reported any correlations among CTC, ctDNA, and exosome and the mutational status of thyroid cancer tissues (es BRAF, RAS, RET/PTC…)? What are the considerations of the authors regarding these correlations??
· In the Abstract, authors reported:” liquid biopsy has been shown to be critical in the thyroid cancers’ diagnosis, treatment, and prognosis in numerous trials”. Thus, in the manuscript the authors should add a novel table regarding the trials in with the liquid biopsy/thyroid cancer is reported.
· Finally, in the Introduction and Discussion sections, it would be interesting if the authors report which is the most studied cancer for the application of liquid biopsy.
-The paragraph 4.2.1 and paragraph 4.2.2. need to be schematized in two tables.
- Please correct the Title of 4.2 Paragraph
Minor editing of English language is required
Reviewer 3 Report
The review provides an overview on the application of liquid biopsy, specifically circulating tumor DNA (ctDNA), circulating tumor cells (CTCs), and exosomes, in the diagnosis, treatment, and prognosis of thyroid cancer. The author highlights the importance of addressing urgent issues in thyroid cancer management and how liquid biopsy may offer a non-invasive and comprehensive approach to solve these challenges. The review also mentions the potential clinical applications of liquid biopsy in early detection, therapy monitoring, and prognostic prediction.
1) line 28-30: the statement is not very clear. Restructuring the sentence might help showing the meaning the author is trying to convey.
2) line 45, page 2: please add an abbreviation to 131I or indicate before it that it’s radioactive Iodine.
3) Figure 1: it is hard to follow the sequence of the diagrams without numbering/labeling. Kindly label/number the three diagrams in figure 1 and add a description in the figure legend for each part. The figure is nice, but without enough detailed description in the figure legend the message is lost.
4) In figure 2: the way the colored boxes (early diagnosis, effective monitoring, precise treatment, and prognostic prediction) are located is confusing. It seems as though “early diagnosis” is only for ctDNA and effective monitoring is only for exosomes, etc….). For example, please put the boxes all together on the rights side with two arrows coming covering the diagram and going towards the boxes. Also add a title for the boxes to indicate that these are “potential tools for:” or if you give a better title it’s also ok.
5) line 96: “EMT may play an critical role in CTCs”, remove the n from an.
6) line 109, page 4: “electrophoretic separation” is mentioned without any explanation what it is. It’s also not included in the
7) line 110, page 4: “Physical-based enrichment methods are simple” please add between parenthesis what those physical-based methods are.
8) line 114, page 4: explain briefly maybe between parenthesis what EpCAM is.
9) Figure 3. Please also add detailed description in the figure legend.
10) Table 1: the separation between CTC, ctDNA and exosomes is not very clear. It appears as though qPCR and ddPCR belong to the CTC section. Please format the table to have a clearer separation between sections.
11) line 176, page 6: there is a letter t at the end of the line that I think should not be there?
12) line 177, page 6: Radioiodine therapy is suddenly written as I131 when is was previously written as “131I”. please be consistent with the format that you want to use.
13) line 205, page 7: is “future therapeutic techniques” an accurate statement when reference 71 is about “Advances in early detection methods for solid tumors”? shouldn’t it be “future diagnostic techniques”? or something along those lines?
14) line 230-231, page 7: I don’t think it’s correct to write “a few research”. Instead write: “a few research studies”.
15) table 3: please add lines between the different studies to have a clearer separation.
16) line 240-241, page 8: the entire sentence is unclear. Please reformulate.
17) line 248, page 9: remove either one of “demonstrated found”.
18) line 255, page 9: add studies after “in several research”
19) line 256, page 9: add abbreviation for CpG.
20) line 258, page 9: add abbreviation for MGMT
21) line 263, page 9: “and seven (70%) were positive”. Is it 7 or 70%?
22) line 311, page 10: why is it more environmentally friendly? This is not discussed.
23) line 317, page 10: “Exosomes in thyroid cancerz” remove the z from cancerz
24) line 328, page 10: fix resum to serum.
25) line 341, page 10: fix lumph to lymph.
26) table 4: also add a separation line between the different studies
27) line 353, page 12: sentence is unclear/incorrect. Please fix
28) line 391, page 13: change levles to levels and add be before “associated”
29) line 398, page 13: change rerum to serum and remove the “and” before “revealing”
30) line 411, page 13: add abbreviation for ATC
31) line 422, page 14: change lackhe to lack.
32) I noticed that the author doesn’t use et al. when citing references in text. Is that not a requirement of the journal?
33) line 429-430, page 14: “Studies have shown that using multiple biomarkers in combination to predict cancer holds more predictive power than using single biomarkers alone.” Are there any studies which try to use the combination of these biomarkers together? Even if or other types of tumors, it would be nice to include such studies.
English language is good in general, except for some minor mistakes which I have indicated in my main report.
Round 2
Reviewer 2 Report
the authors have exhaustively answered all questions. In my opinion, this uploaded manuscript is now suitable for the publication.